# Proposal for a shared definition of « primary healthcare » by health professionals: A national cross-sectional survey

Michel Prade[1]*, Anne Rousseau[2,3,4], Olivier Saint-Lary[1,5], Sophie Baumann[2], Louise Devillers[1,5], Arnaud Courtin[1], Sylvain Gautier[5,6,7]

1 General Practice Department, University of Versailles-Saint-Quentin-en-Yvelines, Montigny-Le-Bretonneux, France, 2 Midwifery Department, Versailles Saint Quentin University, Montigny-le-Bretonneux, France, 3 Department of Obstetrics and Gynecology, Poissy-Saint Germain Hospital, Poissy, France, 4 CESP, UVSQ, INSERM U1018, Clinical Epidemiology Team, University Paris-Saclay, Villejuif, France, 5 CESP, UVSQ, INSERM U1018, Primary Care and Prevention Team, University Paris-Saclay, Villejuif, France, 6 Hospital Department of Epidemiology and Public Health, Raymond Poincaré Hospital, GHU Paris Saclay University, Assistance Publique—Hôpitaux de Paris, Garches, France, 7 University Department "Public Health, Prevention, Observation, Territories", Versailles Saint-Quentin University, Montigny-Le-Bretonneux, France

* michel_prade@hotmail.fr

**Data Availability Statement:** All relevant data are within the paper and its Supporting Information files.

## Abstract

### Introduction

Forty years passed between the two most important definitions of primary health care from Alma Alta Conference in 1978 to WHO's definition in 2018. Since then, reforms of healthcare systems, changes in ambulatory sector and COVID 19, have created a need for reinterpretations and redefinition of primary healthcare. The primary objective of the study was to precise the definitions and the representations of primary healthcare by healthcare professionals.

### Methods

We conducted a descriptive cross-sectional study using a web-based anonymized questionnaire including opened-ended and closed-ended questions but also "real-life" case-vignettes to assess participant's perception of primary healthcare, from September to December 2020. Five case-vignette, describing situations involving a specific primary health care professional in a particular place for a determined task were selected, before the study, by test/retest method.

### Results

A total of 585 healthcare practitioners were included in the study, 29% were general practitioners and 32% were midwives. Amongst proposed healthcare professions, general practitioners (97.6%), nurses (85.3%), midwives (85.2%) and pharmacists (79.3%) were those most associated with primary healthcare. The functions most associated with primary healthcare, with over 90% of approval were "prevention, screening", "education to good

**Funding:** The authors received no specific funding for this work.

**Competing interests:** The authors have declared that no competing interests exist.

health", "orientation in health system". Two case-vignettes strongly emerged as describing a situation of primary healthcare: Midwife/Hospital/Pregnancy (74%) and Pharmacist/Pharmacy/Flu shot (90%). The profession and the modality of practice of the responders lead to diverging answers regarding their primary healthcare representations.

## Conclusions

Primary healthcare is an ever-evolving part of the healthcare system, as is its definition. This study explored the perception of primary healthcare by French healthcare practitioners in two complementary ways: oriented way for the important functions and more practical way with the case-vignettes. Understanding their differences of representation, according to their profession and practice offered the authors a first step to a shared and operational version of the primary healthcare definition.

## Introduction

The concept of "primary health care" (PHC) was first defined in 1978 at the Alma-Ata conference as "essential health care, economically and socially sustainable" [1]. A more recent definition was proposed, forty years later during the Astana conference in 2018: « PHC is a whole-of-society approach to health that aims equitably to maximize the level and distribution of health and well-being by focusing on people's needs and preferences (both as individuals and communities) as early as possible along the continuum from health promotion and disease prevention to treatment, rehabilitation and palliative care, and as close as feasible to people's everyday environment » [2].

In the 40 years between these two definitions, the concept of PHC has been reinterpreted and redefined many times, producing confusion in both terminology and practice. Barbara Starfield, at the end of the 1990s, proposed a definition of PHC centered on the activity of the general practitioner (GP) and introduced 4 essential functions: "first contact", "longitudinality (person-focused care over time)", "comprehensiveness", "coordination". Starfield has also demonstrated that health systems based on PHC are more efficient and less costly [3, 4]. Based on these publications, in most OECD countries [5], the concept of PHC has been integrated to refer to the first segment of the health system. Ambulatory medicine, its structures, and actors, have thus been able to identify with this concept of PHC. In France, the health system is historically divided between a hospital sector, largely dominated by public institutions, and an ambulatory sector in which private practice, partly financed by public funding, predominates. The concept of PHC has been used to describe this ambulatory care sector and its professionals [6]. Gradually, the concept has become accepted by health care actors until it has been incorporated into the law [7].

However, over the last twenty years, the sustained reforms of the French health system and of the ambulatory sector in particular, have strongly reexamined the outlines of PHC [8, 9]. These reforms have upset the representations of some and others and modified professional positions. They have redistributed the roles of many professionals. Advanced practice nurses are a significant example, during these successive reforms they gained more autonomy in their practice and are becoming a significant part of PHC [10, 11]. Pharmacists are another profession concerned by these redistributions of roles. In France, they are now able to vaccinate against the flu and were able to vaccinate during the COVID-19 pandemic [12]. Other

professionals could perceive a shift in what they were expected to do. In France, general practitioners are strongly associated with the gatekeeping of the primary healthcare sector [13]. Recent reforms have emphasized this prerogative, the general practitioner having a central role in coordination of patient healthcare and orientation in the health system. Those changes institute a blur in the nature and prerogatives of PHC professionals and raise questions about a revision of the definition of primary healthcare.

On a territorial scale, the recent COVID-19 pandemic has only accentuated these questions, at a time when new large-scale reforms are expected. Some people see the reforms on the hospital sector and the emergence of "local hospitals" as an opportunity to reconsider the roles and functions of the various actors on a territorial scale. This situation is common in many countries in which epidemiological and health transitions lead to a rethinking of the organization and structuring of the health system in order to meet the challenges of aging, chronic diseases, increased mobility, and emerging diseases [14]. To better communicate and organize the health system on a territorial level, it is significant for the professionals to understand that they belong to a sector of this system. This is especially needed in a moving, subject to reforms system. It is important to comprehend the other professional's prerogatives and abilities to develop teamwork and interprofessional collaboration [15]. Interprofessional collaboration is essential to provide efficient and quality patient care [16, 17]. In order to develop the collaboration between health care actors, they have to "speak the same language" [18].

The apparent changes in the outlines of PHC and their translation into a given health system now invites us to question the perception of this concept by health professionals. To clarify the definitions and representations of PHC seems essential to better allocate human and financial resources that will be deployed in future reforms of the health system. The primary objective of our study was to precise the definitions and the representations of primary healthcare by healthcare professionals. The secondary objective was to identify the determinants associated with the different representations.

## Material and methods

### Study design

This study is a descriptive cross-sectional study using a web-based anonymized questionnaire. We used closed-end questions, opened ended questions and vignettes-based questionnaire to assess the participant's perception of primary healthcare. The study took place from 09/27/2020 to 12/02/2020.

### Participants

The respondents included in the study were French healthcare workers still in practice who responded to the full survey. The respondents were excluded from the study if they didn't finish the survey, if they already answered the survey once, if they weren't healthcare workers or if they weren't still practicing. We tried to reach as many professionals as possible to be able to study the most diverse population in term of profession, activity modalities and university involvement. In pursuit of that goal we chose to broadcast the survey to a selection of mailing lists known for their high response rate. We also recruited responders in social media, used publicly accessible email addresses of different professionals in all of France. The diffusion strategy also relied on the snowball effect, as the respondents were encouraged to distribute the questionnaire to their contact list.

Respondents' informed consent was sought through an email invitation which contained the survey link, an explanation of the study's purpose and its identity protection measures.

Consenting invitees could immediately participate. The study was open for data collection for eleven weeks, during which gentle reminder emails were sent to non-responders.

## Survey instrument

The study used the opensource software *Limesurvey* to build, host and broadcast the survey. We build the survey using the "Checklist for Reporting Results of Internet E-Surveys" (CHERRIES) guidelines. The survey consisted of three components, for an estimated completion time of less than ten minutes. To avoid multiple completion, each participant was asked if he had already completed the survey once. If the answer was positive, the survey stopped automatically, and the answer was discarded. It was impossible to go back when completing the survey.

The first component was meant to describe the population's characteristics. We asked for: Age, Gender, Profession, Modalities of their practice (liberal, city or hospital. . .), Degree of involvement in university (rated from 1 = No involvement to 4 = Frequent involvement in university)

The second component of the questionnaire consisted of an opened ended question and multiple-choice questions to retrieve the participant's naive definition of primary healthcare. Firstly, respondents were asked by an open-ended question to suggest three key words describing what PHC meant to them. Secondly, in multiple choice questions, respondents had to choose amongst a list of functions, healthcare workers and places linked to PHC in literature [3, 6], those who they thought could be a part of their definition of PHC.

The third component of the survey consisted of 5 standardized clinical case-vignette placing the respondents in "real life" situation. Vignette studies has been used by many authors in medical literature [19, 20]. We used this specific method to capture the respondent's judgement or inclination to categorize a situation towards primary healthcare. Each of the selected clinical scenarios staged a different place, healthcare professional and function of care. We selected the five case-vignettes after a test/retest of the vignette method, taking place for 3 months, before the study started. The test/retest consisted of a survey composed of 15 vignettes, submitted to 50 healthcare professionals. They were asked for each vignette if the situation could be considered as a situation of primary healthcare or not. Three months later, we asked them to respond to the exact same survey. The test/retest showed there was no significant divergence between the two surveys after a 3 months interval. For each case-vignette, the Kappa score was between 0.62 and 0.92, displaying a strong agreement. We concluded that there was a good chance we could capture the respondent's clear opinion with a case-vignette method. Amongst the 15 vignettes tested, we chose the 5 final case-vignettes on the basis of their Kappa score, which would be the highest possible and the variety of the situations pictured in each case.

## Statistical analysis

Characteristics of participants were described with frequencies and percentages for qualitative variable.

Regarding opened ended question, two of the co-authors (AR and MP) conducted the content analysis of the open-ended questions. They classified responses by themes in 14 categories and resolved any disagreement in consultation with a third author (SG). Responses categories were then described by their frequencies and percentages.

We performed descriptive analysis on the Multiple-Choice Questions aimed to clarify the definition of primary healthcare by the respondents. For each clinical case-vignettes, we did a descriptive analysis and statistical tests on every explicative variable. Dependent variables were case-vignettes considered as primary healthcare and explicative variables were the age, the sex,

the profession and the degree of university involvement. Qualitative variables were analyzed with the Chi-square test or Fisher's test, as appropriate. All statistical tests were two-sided, and $p<0.05$ was considered statistically significant. Statistical analysis was conducted with R version 4.0.

### Ethics approval

There were no ethical issues raised upon the realization of the study. We consulted the ethics comity of the national organization of teaching general practitioner (IRB number 00010804). Anonymity was guaranteed in the study. No personal data was collected. We registered the study to the "CNIL" (National comity of computer science and liberties), a French independent administration in charge of informatic data protection (Registering number 2217819). Participant's consent was obtained at the start of the survey, consent could be waived at any moment during the study.

## Results

### Characteristics

A total of 585 participants were included in the study (Fig 1).

Participant's characteristics are displayed in Table 1. Most of the participants were women (75.7%), most represented professions were General Practitioners (29.1%) and midwifes (32.5%). About half of the participants were working in hospitals, and about half of the participants were involved in university work.

### Naïve perception of primary healthcare

The study gathered 1401 key words from the open-ended question, then distributed in 14 categories (Table 2). The most represented categories when asked about PHC were "prevention, screening, education to a good health" (44.3%), "Accessibility and proximity" (33.8%) and "first resort, gatekeeping" (26.2%).

### Professionals and places associated with primary healthcare

Amongst proposed healthcare professions, GP (97.6%), nurse (85.3%), Midwife (85.2%) and pharmacist (79.3%) were those most associated with primary healthcare. At the opposite, radiologist (18.8%), cardiologist (21.7%), paramedic (21.9%) and psychiatrist (27.5%) were the professions least associated with primary healthcare (Fig 2). When asked if their profession was associated with primary healthcare, 100% of GPs answered yes, 98.9% of nurses and 98.9% of midwives. Amongst proposed places of healthcare, General practitioner's office (94.2%), multi-professional medical practice (93.5%), Maternal and child protection office (85.5%), Mental health medical center (63.1%) were those most associated with primary healthcare.

### Adherence to the functions of primary healthcare cited in literature and French law

More than 50% of the respondents mostly agrees or totally agrees with the functions cited in literature and French law as prerogatives of primary healthcare (Fig 3). The functions most associated with primary healthcare, with over 90% of approval are "prevention, screening", "education to good health", "orientation in health system".

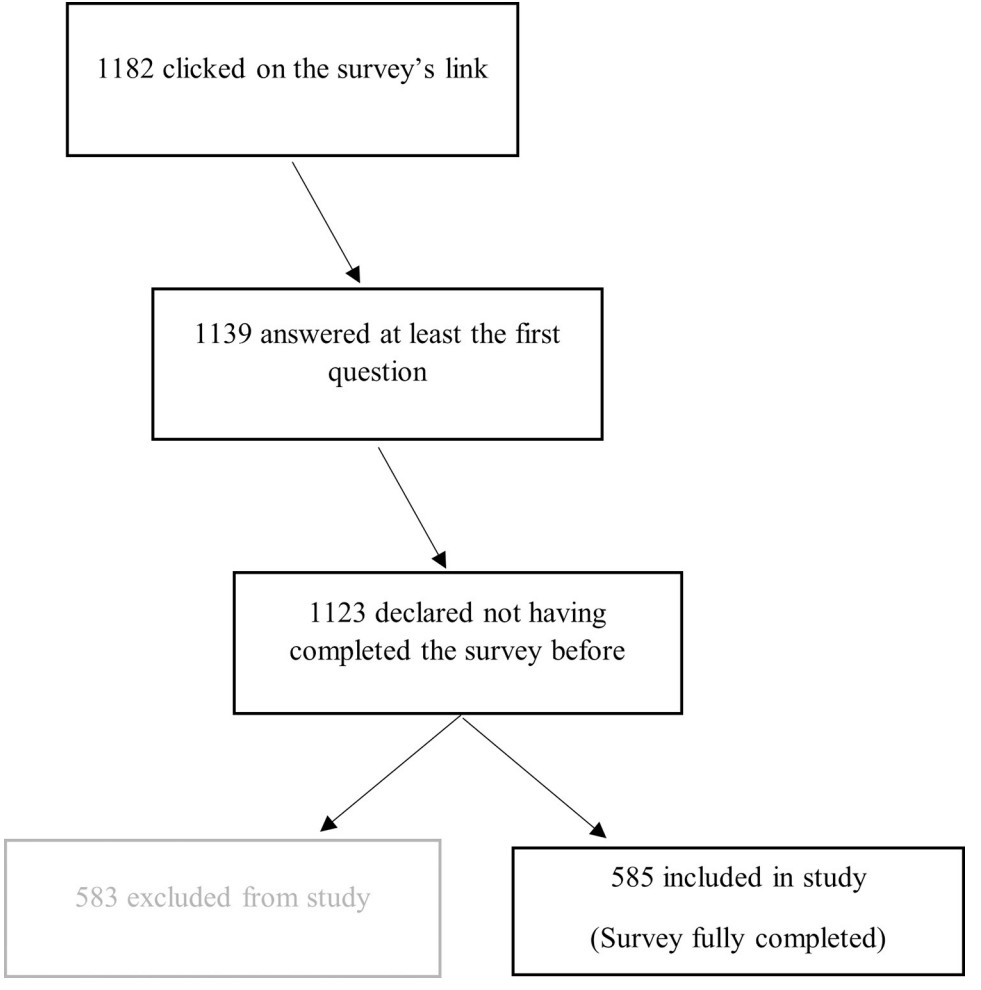

**Fig 1. Flow chart.**

## Case vignettes

The 5 case-vignettes implemented in the survey are displayed in Table 3. Most participants thought that the first two vignettes described situation of primary healthcare: Midwife/Hospital/Pregnancy (74%) and Pharmacist/Pharmacy/Flu shot (90%) (Table 4).

Regarding potential determinants, the profession of the responders was a variable associated with a divergence of perception in four vignettes: Midwife/Hospital/Pregnancy ($p<0.01$), Cardiologist/Practice/EKG ($p<0.001$), Occupational Physician/Company/Fracture ($p<0.01$), Nurse/Emergency Room/Abdominal pain ($p<0.01$). The modalities of practice of the responders were explored using a variable associated with a divergence of perception in two vignettes: Midwife/Hospital/Pregnancy ($p<0.01$) Occupational Physician/Company/Fracture ($p<0.001$). The university involvement of the responders was a variable associated with a divergence of perception in two vignettes: Midwife/Hospital/Pregnancy ($p = 0.03$) Cardiologist/Practice/EKG ($p<0.001$) (Table 4).

## Discussion

### Principal findings of the study

The study highlighted places, professionals and functions strongly associated with the perception of primary healthcare by the respondents. Together, they formed a shared definition of

**Table 1. Characteristics of population.**

| Characteristics | N = 585 |
|---|---|
| | n (%) |
| | mean [minimum; maximum] |
| Women | 443 (75.7) |
| Men | 142 (24.3) |
| Age | 40.6 [21; 80] |
| Midwife | 189 (32.5) |
| General practitioner | 167 (29.1) |
| Other practitioner | 31 (5.3) |
| Nurse | 89 (15.3) |
| Physiotherapist | 6 (1.0) |
| Pharmacist | 65 (11.2) |
| Student | 11 (1.9) |
| Dentist | 5 (0.9) |
| Other paramedical | 16 (2.8) |
| Liberal practice | 279 (47.7) |
| Hospital practice | 306 (52.3) |
| None or few involvements in university work | 308 (52.6) |
| Involved in university work | 277 (47.4) |

primary healthcare based on the missions of prevention, education to a good health, follow up of patient's health and gate keeping.

The case-vignette study demonstrated variations in the perception of what is primary healthcare in French context, according to healthcare professionals. Those divergence were depending on the characteristics of the healthcare professionals we questioned.

**Table 2. Classification of key words from open–ended question.**

| Category | Number of key words classified in the category | % |
|---|---|---|
| Prevention, screening, education to good health | 259 | 44.3 |
| Accessibility, proximity | 198 | 33.8 |
| First resort, gatekeeping | 153 | 26.2 |
| Diagnostic, treatment, follow up | 129 | 22.1 |
| Orientation in system of care | 109 | 18.6 |
| Global patient care | 103 | 17.6 |
| Quality of the healthcare professional | 87 | 14.9 |
| Essential care | 87 | 14.9 |
| Public health | 53 | 9.1 |
| General practitioner | 50 | 8.5 |
| Unscheduled care, emergency | 50 | 8.5 |
| Ambulatory, liberal | 44 | 7.5 |
| Universal | 43 | 7.4 |
| Life and death | 36 | 6.2 |

Key words gathered from the open–ended question: "What does primary healthcare mean to you?" classified in 14 categories

| Professionals | Full sample n (%) | GP | Midwife | Nurse | Pharmacist |
|---|---|---|---|---|---|
| General practitioner (GP) | 571 (97.6) | 169 (100) | 183 (96.8) | 86 (96.6) | 62 (95.4) |
| Nurse | 499 (85.3) | 145 (85.8) | 152 (80.4) | 88 (98.9) | 58 (89.2) |
| Midwife | 487 (83.2) | 132 (78.1) | 187 (98.9) | 68 (76.4) | 47 (72.3) |
| Pharmacist | 464 (79.3) | 145 (85.8) | 134 (70.9) | 67 (75.3) | 62 (95.4) |
| Dentist | 407 (69.6) | 127 (75.1) | 138 (73.0) | 53 (59.6) | 41 (63.1) |
| Pediatrician | 373 (63.8) | 111 (65.7) | 120 (63.5) | 56 (62.9) | 38 (58.5) |
| Home Help | 333 (56.9) | 103 (60.9) | 98 (51.9) | 57 (64.0) | 36 (55.4) |
| Occupational physician | 331 (56.6) | 65 (38.5) | 132 (69.8) | 54 (60.7) | 36 (55.4) |
| Physiotherapist | 326 (55.7) | 106 (62.7) | 95 (50.3) | 52 (58.4) | 35 (53.8) |
| Psychologist | 283 (48.4) | 90 (53.3) | 82 (43.4) | 52 (58.4) | 24 (36.9) |
| Social worker | 258 (44.1) | 93 (55.0) | 69 (36.5) | 40 (44.9) | 28 (43.1) |
| Emergency physician | 218 (37.3) | 61 (36.1) | 66 (34.9) | 22 (24.7) | 33 (50.8) |
| Medical secretary | 195 (33.3) | 86 (50.9) | 46 (24.3) | 28 (31.5) | 17 (26.2) |
| Ophthalmologist | 161 (27.5) | 34 (20.1) | 59 (31.2) | 28 (31.5) | 20 (30.8) |
| Psychiatrist | 161 (27.5) | 45 (26.6) | 41 (21.7) | 28 (31.5) | 22 (33.8) |
| Paramedic | 128 (21.9) | 41 (24.3) | 31 (16.4) | 20 (22.5) | 18 (27.7) |
| Cardiologist | 127 (21.7) | 17 (10.1) | 44 (23.3) | 29 (32.6) | 21 (32.3) |
| Radiologist | 110 (18.8) | 28 (16.6) | 36 (19.0) | 17 (19.1) | 16 (24.6) |
| **Places** | **Full sample n (%)** | **GP** | **Midwife** | **Nurse** | **Pharmacist** |
| General practitioner's office | 551 (94.2) | 169 (100) | 175 (92.6) | 83 (93.3) | 57 (87.7) |
| multi-professional medical practice | 547 (93.5) | 161 (95.3) | 179 (94.7) | 82 (92.1) | 60 (92.3) |
| Maternal and child protection office | 500 (85.5) | 148 (87.6) | 171 (90.5) | 74 (83.1) | 46 (70.8) |
| Mental health medical center | 369 (63.1) | 108 (63.9) | 128 (67.7) | 50 (56.2) | 41 (63.1) |
| Prison health service | 364 (62.2) | 105 (62.1) | 126 (66.7) | 55 (61.8) | 32 (49.2) |
| Maternity Clinic | 339 (57.9) | 67 (39.6) | 161 (85.2) | 40 (44.9) | 32 (49.2) |
| « On call » general practitioner's office | 315 (53.8) | 115 (68.0) | 83 (43.9) | 40 (44.9) | 41 (63.1) |
| Reeducation center | 232 (39.7) | 78 (46.2) | 71 (37.6) | 31 (34.8) | 26 (40.0) |
| Emergency ward | 231 (39.5) | 63 (37.3) | 73 (38.6) | 28 (31.5) | 31 (47.7) |
| Laboratory | 187 (32.0) | 71 (42.0) | 46 (24.3) | 23 (25.8) | 26 (40.0) |
| Radiology center | 123 (21.0) | 39 (23.1) | 35 (18.5) | 14 (15.7) | 19 (29.2) |

| 0% - 29% | 30% - 49% | 50% - 79% | 80% - 100% |
|---|---|---|---|
|  |  |  |  |

**Fig 2. Professionals and places associated with primary healthcare.**

## Strengths and limitations

The study was based on a validated methodology, especially the case-vignette part, a method extensively studied in literature and used to collect the respondent's perception [9, 10]. To strengthen the methodology, selected case-vignette endured a session of back test three months prior to the study. The results we obtained were consistent with data gathered from the literature. Core concepts of primary healthcare came out with strength in the study, giving that much value to the rest of the observations.

However, our survey presents some bias. Due to the method of diffusion of the questionnaire, the study was exposed to a selection bias. The highlighted divergence in the respondent's definition and perception of primary healthcare could have been minored by this bias. As a questionnaire based study, the respondents could have been influenced by a social desirability bias. Using case-vignette, we tried to minor the impact of steered responses obtained from a questionnaire based study. The concept of primary healthcare is relatively new to healthcare professionals and to avoid a declarative bias, it was impossible to go back when the questionnaire started.

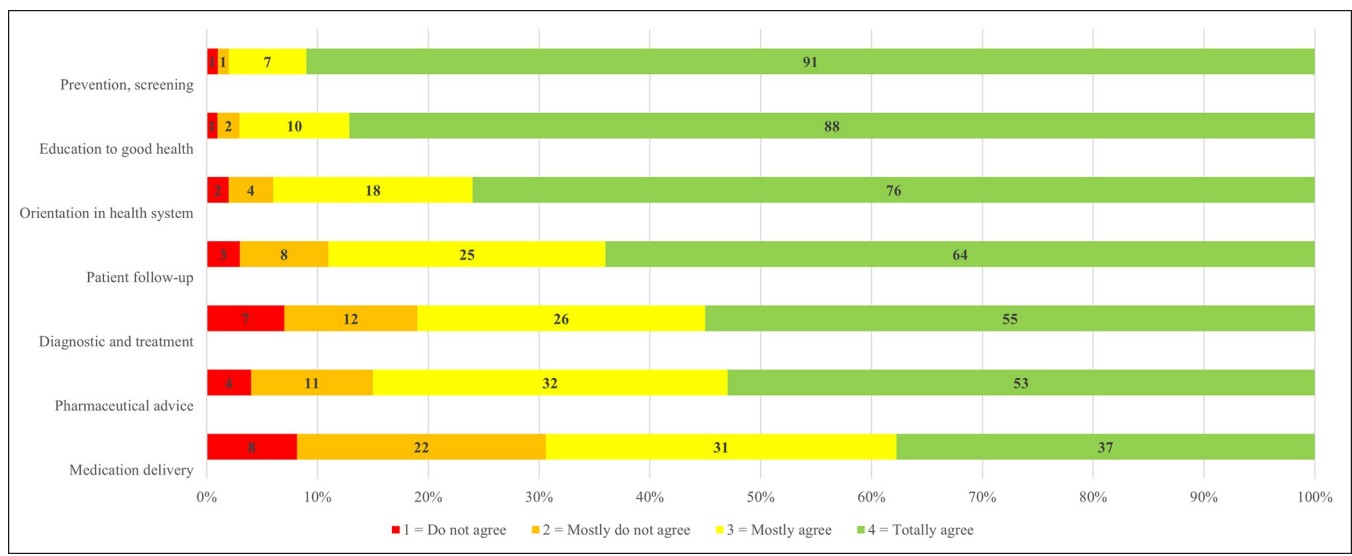

**Fig 3. Functions associated with primary healthcare.**

## Foundations of the primary healthcare definition

Primary healthcare is an ever-evolving part of the healthcare system. From the ideal of social justice described at Alma Ata in 1978, the reaffirmed importance of PHC in a changing world at Astana in 2018, to the necessary, large scale reorganization of healthcare systems due to covid 19, primary healthcare has shown a great capability of adaptation [1, 21]. This adaptability is possible thanks to solid foundations, which form the basis of a definition of PHC.

The idea of social justice was at the core of the PHC definition in the Alma-Ata conference. Our study illustrated this bond with accessibility and proximity being strongly attached to PHC. Prevention, screening, education to a good health and orientation in healthcare system are cited as a major part of the definition. These functions were key points of interest in the Alma Ata conference, and B. Starfield publications [1, 4]. They were reaffirmed in French law and Astana in 2018 and should remain at the center of future evolutions of healthcare policy [9, 21]. These powerful concepts carried by healthcare workers and their places of care form the foundations of PHC.

## A shared definition of primary healthcare

Having a shared definition of PHC could lead to a better collaboration, coordination and communication between the different actors of the health system. PHC in particular involves many

**Table 3. The five case–vignettes.**

| Case-vignette | Professional | Place | Function |
|---|---|---|---|
| Mrs C, 25 years old, is consulting a midwife, in a teaching hospital, for her 6th month pregnancy checkup. | Midwife | Hospital | Pregnancy |
| Mrs D, 70 years old, has an appointment with her pharmacist, in her neighborhood pharmacy to get her flu shot. | Pharmacist | Pharmacy | Flu shot |
| Mr V, 55 years old, is consulting his usual cardiologist in his practice, to realize an EKG regarding the annual checkup of his high blood pressure. | Cardiologist | Practice | EKG |
| Mr D, 50 years old is consulting the occupational physician in the premises of his company to discuss coming back to work after a tibial fracture during his last ski vacation. | O. Physician | Company | Fracture |
| Mrs P, 34 years old is cared for by an emergency nurse, working in the emergency department of a small hospital. She has important abdominal pain since this morning. | Nurse | Emergency Room | Abdominal pain |

**Table 4. Case–vignette analysis.**

| | Midwife Hospital Pregnancy | P | Pharmacist Pharmacy Flu shot | p | Cardiologist Practice EKG | p | O.physician Company fracture | P | Nurse Emergency Dpt Abdominal pain | P |
|---|---|---|---|---|---|---|---|---|---|---|
| Yes (%) | N = 434 (74.2%) | P | N = 587 (90.1%) | p | N = 275 (47.0%) | p | N = 293 (50.1%) | P | N = 287 (49%) | P |
| **Age** | | 0.94 | | **0.002** | | 0.902 | | 0.557 | | 0.312 |
| <38 years | 73.8 | | **94** | | 42.3 | | 51 | | 51.3 | |
| >38 years | 73.2 | | **86.1** | | 41.4 | | 48.2 | | 46.8 | |
| **Gender** | | < 0.001 | | 0.892 | | 0.208 | | <0.01 | | 0.097 |
| Men | **62.7** | | 89.4 | | 37.3 | | **40.1** | | 55.6 | |
| Women | **77.2** | | 90.3 | | 43.8 | | **53.3** | | 47.2 | |
| **Profession** | | < 0.01 | | 0.441 | | <0.001 | | <0.01 | | <0.01 |
| Midwife | **89.4** | | 91.5 | | **39.2** | | **56.1** | | **40.2** | |
| General practitioner | **64.5** | | 89.3 | | **35.5** | | **40.2** | | **53.8** | |
| Nurse | **71.9** | | 85.4 | | **64** | | **58.4** | | **46.1** | |
| Pharmacist | **61.5** | | 89.9 | | **38.5** | | **56.9** | | **61.5** | |
| Other profession | **65.2** | | 93.8 | | **40.6** | | **37.7** | | **56.5** | |
| **Modalities of practice** | | <0.01 | | 0.94 | | 0.103 | | <0.01 | | 0.191 |
| Liberal (solo) | **65.8** | | 88.8 | | 41.4 | | **40.1** | | 48.7 | |
| Liberal (pluriprofessional) | **63.8** | | 90.6 | | 33.9 | | **46.5** | | 44.9 | |
| Clinic or hospital | **81.8** | | 90.9 | | 49.1 | | **50** | | 58.2 | |
| Other type of practice | **81.6** | | 90.3 | | 44.4 | | **60.2** | | 47.4 | |
| **University involvement** | | 0.03 | | 0.722 | | <0.001 | | 0.139 | | 0.283 |
| 1 = None | **71.3** | | 89.3 | | **47.5** | | 51.6 | | 56.6 | |
| 2 = Few | **78** | | 90.3 | | **46.8** | | 52.2 | | 47.3 | |
| 3 = Regular | **77** | | 88.8 | | **44.4** | | 52.8 | | 48.9 | |
| 4 = Very frequently | **62.6** | | 92.9 | | **23.2** | | 39.4 | | 44.4 | |

actors, caregivers or administrative and organizational staff. To better work together, they have to understand each other's roles and functions. This improved collaboration in PHC has to be based on a common ground, a shared definition where the field of PHC and every actor's prerogatives are clearly stated. A shared definition of PHC could be a path to promote inter-professional communication and collaboration. In a 2015 publication by *Supper et.al*, it is said "In the early stages of collaboration, time should be dedicated to communication, training, building shared views and overcoming prejudices, to save time later on" and "The main barriers were the challenges of definition and awareness of one another's roles and competences." [22]. With an improved collaboration between professionals, studies has shown a positive impact on patient's health [23, 24]. We think a better understanding of the prerogatives of the healthcare workers composing PHC and the boundaries of said system can lead to a better allocation of human and financial resources. On the contrary, a poor understanding of what is PHC could lead to an overlap of functions between different professionals, causing misunderstanding.

While specifying the outlines of PHC and trying to better collaboration between their actors, having a shared definition of PHC can bring challenges. One of them would be to crystallize the partition between primary and secondary care. In our study, the respondents

described PHC in opposition to the hospital centered side of the healthcare system. This is found in Bismarckian rooted social protection systems where primary healthcare builds itself in opposition to public health institutions [25, 26]. With a clearly stated definition, those distinctions could lead to a compartmentalization of the healthcare system, giving strength to the identity of PHC and secondary healthcare but making communication between both of the entities less efficient [27]. The relationship between general practitioner and hospital-based specialists can be used to illustrate the compartmentalization between PHC and a hospital centered secondary healthcare. Sometimes described as opposites of each other, they are nevertheless responsible for the continuity of the patient's care [25]. GPs and medical specialists are two parts of the healthcare system in constant communication around the patient, in an attempt to provide the most effective care possible [27, 28]. The two parts not being from the same healthcare sector, and their priorities being different, the expectations in term of communication may vary, causing friction between the actors, further indicating the opposition between them [28, 29].

## Conclusion

Pieces of an operational definition of PHC has emerged from this study. Concepts strongly associated with PHC were highlighted and the perception of PHC by his effectors, healthcare professionals helped to understand recent changes. A shared definition of PHC can foster interprofessional collaboration and communication between professionals. On a territorial scale, it could help for a better allocation of the resources available. A shared definition also poses challenges. Primary care and its prerogatives being strengthened, the risk is to reinforce the partition between primary and secondary care.

To further understand and enrich the proposed definition of PHC, we chose to conduct in extension to this study a qualitative study. The objective was, with the contribution of the quantitative study to try and materialize an operational version of the PHC definition.

## Supporting information

**S1 Table. Database of healthcare workers responses.**
(XLSX)

## Author Contributions

**Conceptualization:** Michel Prade, Anne Rousseau, Olivier Saint-Lary, Sophie Baumann, Arnaud Courtin, Sylvain Gautier.

**Formal analysis:** Michel Prade, Sylvain Gautier.

**Funding acquisition:** Anne Rousseau.

**Investigation:** Michel Prade, Anne Rousseau, Arnaud Courtin, Sylvain Gautier.

**Methodology:** Michel Prade, Anne Rousseau, Olivier Saint-Lary, Sophie Baumann, Arnaud Courtin, Sylvain Gautier.

**Project administration:** Anne Rousseau, Sylvain Gautier.

**Supervision:** Michel Prade, Anne Rousseau, Sylvain Gautier.

**Validation:** Anne Rousseau, Olivier Saint-Lary, Sylvain Gautier.

**Writing – original draft:** Michel Prade.

**Writing – review & editing:** Michel Prade, Anne Rousseau, Olivier Saint-Lary, Sophie Baumann, Louise Devillers, Sylvain Gautier.

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
