## [Editor Report · Decision Letter 0]

13 Dec 2022

PONE-D-22-28991Proposal for a shared definition of « primary healthcare » by health professionals: a national cross-sectional surveyPLOS ONE

Dear Dr. Prade

Thank you for submitting your manuscript to PLOS ONE. After careful consideration, we feel that it has merit but does not fully meet PLOS ONE’s publication criteria as it currently stands. Therefore, we invite you to submit a revised version of the manuscript that addresses the points raised during the review process.

This paper surveys health professionals to determine what definitions of primary health care they share or do not share. This is a valuable study to see how primary health care, which has received a lot of attention after the Alma Ata Declaration, is perceived by health professionals.

However, more needs to be written about the background of why this study was undertaken. If there is a reality that there is a lack of common understanding among health professionals in promoting primary health care, please write specifically about it as a basis for raising the issue.

In addition, Discussion and Conclusion should be written in more depth. Please analyze the significance of having a common definition, the problems caused by the lack of a shared definition, and what negative impact this may have on healthcare. Also, please provide specifics on what having a common definition of primary health care would entail.

We look forward to receiving the manuscript with further revisions.

We look forward to receiving your revised manuscript.

Kind regards,

Miwako Hosoda

Academic Editor

PLOS ONE

Journal Requirements:

When submitting your revision, we need you to address these additional requirements. 1. Please ensure that your manuscript meets PLOS ONE's style requirements, including those for file naming. The PLOS ONE style templates can be found at https://journals.plos.org/plosone/s/file?id=wjVg/PLOSOne_formatting_sample_main_body.pdf and https://journals.plos.org/plosone/s/file?id=ba62/PLOSOne_formatting_sample_title_authors_affiliations.pdf 2. Please note that the supplementary file "Database.xlsx" appears to contain potentially identifying information (i.e. email addresses). These should be anonymised as this information may breach patient confidentiality and these details should not be included. 3. Please provide additional details regarding participant consent. In the ethics statement in the Methods and online submission information, please ensure that you have specified what type you obtained (for instance, written or verbal, and if verbal, how it was documented and witnessed). If your study included minors, state whether you obtained consent from parents or guardians. If the need for consent was waived by the ethics committee, please include this information. 4. Please include captions for your Supporting Information files at the end of your manuscript, and update any in-text citations to match accordingly. Please see our Supporting Information guidelines for more information: http://journals.plos.org/plosone/s/supporting-information.****

---

## [Author Response · Author response to Decision Letter 0]

26 Jan 2023

Dear Editor, 

We thank you for the response received on December 14, concerning our manuscript entitled “Proposal for a shared definition of « primary healthcare » by health professionals: a national cross-sectional survey” by Michel Prade, Anne Rousseau, Olivier Saint-Lary, Sophie Baumann, Louise Devillers, Arnaud Courtin, Sylvain Gautier, informing us you would be willing to give further consideration to a revised version. 

The authors are very grateful to the Editor for her constructive help. We think the paper has been much improved. Our revised version has taken into account all of the points raised, as detailed below and in the rebutal letter. 

Comments to the Author : 

However, more needs to be written about the background of why this study was undertaken. If there is a reality that there is a lack of common understanding among health professionals in promoting primary health care, please write specifically about it as a basis for raising the issue.

Author’s response to comment: 

We thank the editor for this suggestion, which encourages us to better present the context of why our study was undertaken. We have added examples of redistribution or redefinition of PHC worker’s prerogatives in the introduction section, page 3 and 4, beginning line 84 of the manuscript. These changes could lead to a lack of common understanding among health professionals. “These reforms have upset the representations of some and others and modified professional positions. They have redistributed the roles of many professionals. Advanced practice nurses are a significant example, during these successive reforms they gained more autonomy in their practice and are becoming a significant part of PHC [10,11]. Pharmacists are another profession concerned by these redistributions of roles. In France, they are now able to vaccinate against the flu and were able to vaccinate during the COVID-19 pandemic [12]. Other professionals could perceive a shift in what they were expected to do. In France, general practitioners are strongly associated with the gatekeeping of the primary healthcare sector [13]. Recent reforms have emphasized this prerogative, the general practitioner having a central role in coordination of patient healthcare and orientation in the health system. Those changes institute a blur in the nature and prerogatives of PHC professionals and raise questions about a revision of the definition of primary healthcare.”

 We also added precisions about why a lack of common understanding could lead to issues in the organization of PHC and ultimately patient care in the Introduction section, line 101 as follow: “To better communicate and organize the health system on a territorial level, it is significant for the professionals to understand that they belong to a sector of this system. This is especially needed in a moving, subject to reforms system. It is important to comprehend the other professional’s prerogatives and abilities to develop teamwork and interprofessional collaboration [15]. Interprofessional collaboration is essential to provide efficient and quality patient care [16,17]. In order to develop the collaboration between health care actors, they have to “speak the same language” [18].”

Comments to the Author : 

In addition, Discussion and Conclusion should be written in more depth. Please analyze the significance of having a common definition, the problems caused by the lack of a shared definition, and what negative impact this may have on healthcare. Also, please provide specifics on what having a common definition of primary health care would entail.

We thank the editor for this comment, giving us a chance to develop our discussion and conclusion. We added in the Discussion section, page 11 and 12 beginning line 279, a more in depth analysis of the significance of having a common definition, what having a common definition would entail and the problems caused by the lack of a shared definition. “Having a shared definition of PHC could lead to a better collaboration, coordination and communication between the different actors of the health system. PHC in particular involves many actors, caregivers or administrative and organizational staff. To better work together, they have to understand each other’s roles and functions. This improved collaboration in PHC has to be based on a common ground, a shared definition where the field of PHC and every actor’s prerogatives are clearly stated. A shared definition of PHC could be a path to promote interprofessional communication and collaboration. In a 2015 publication by Supper et.al, it is said “In the early stages of collaboration, time should be dedicated to communication, training, building shared views and overcoming prejudices, to save time later on” and “The main barriers were the challenges of definition and awareness of one another's roles and competences.”[22]. With an improved collaboration between professionals, studies has shown a positive impact on patient’s health [23,24]. We think a better understanding of the prerogatives of the healthcare workers composing PHC and the boundaries of said system can lead to a better allocation of human and financial resources. On the contrary, a poor understanding of what is PHC could lead to an overlap of functions between different professionals, causing misunderstanding.” 

We also added in the Discussion section, page 12 line 293, one of the challenges of having a shared definition of PHC : “While specifying the outlines of PHC and trying to better collaboration between their actors, having a shared definition of PHC can bring challenges. One of them would be to crystallize the partition between primary and secondary care.”

We then developed the Conclusion section, taking into account the suggestions, page 12 L.311 : “A shared definition of PHC can foster interprofessional collaboration and communication between professionals. On a territorial scale, it could help for a better allocation of the resources available. A shared definition also poses challenges. Primary care and its prerogatives being strengthened, the risk is to reinforce the partition between primary and secondary care”

---

## [Editor Report · Decision Letter 1]

2 Feb 2023

Proposal for a shared definition of « primary healthcare » by health professionals: a national cross-sectional survey

PONE-D-22-28991R1

Dear Dr. Michael Prade,

We’re pleased to inform you that your manuscript has been judged scientifically suitable for publication and will be formally accepted for publication once it meets all outstanding technical requirements.

Kind regards,

Miwako Hosoda

Academic Editor

PLOS ONE

Additional Editor Comments (optional):

The revised manuscript showed evidence of a sincere response to my previous comments. Since some modification has been taken, I have decided that this paper is acceptable for publication.
---

## [Editor Report · Acceptance letter]

17 Feb 2023

PONE-D-22-28991R1 

Proposal for a shared definition of « primary healthcare » by health professionals: a national cross-sectional survey 

Dear Dr. Prade:

I'm pleased to inform you that your manuscript has been deemed suitable for publication in PLOS ONE. Congratulations! Your manuscript is now with our production department. 

Kind regards, 

on behalf of

Dr. Miwako Hosoda 

Academic Editor

PLOS ONE